# Examining the Influence of the Human Gut Microbiota on Cognition and Stress: A Systematic Review of the Literature

**DOI:** 10.3390/nu14214623

**Published:** 2022-11-02

**Authors:** Matthew B. Cooke, Sarah Catchlove, Katie L. Tooley

**Affiliations:** 1Department of Health Sciences and Biostatistics, Swinburne University, Melbourne 3122, Australia; 2Centre for Human Psychopharmacology, Swinburne University, Melbourne 3122, Australia; 3Cognition & Behaviour, Land Division (Edinburgh), Defence Science & Technology Group, Department of Defence, Edinburgh 5111, Australia

**Keywords:** microbiota, cognition, stress, anxiety, depression, psychobiotics

## Abstract

The gut microbiota is seen as an emerging biotechnology that can be manipulated to enhance or preserve cognition and physiological outputs of anxiety and depression in clinical conditions. However, the existence of such interactions in healthy young individuals in both non-stressful and stressful environments is unclear. The aim of this systematic review was to examine the relationship between the human gut microbiota, including modulators of the microbiota on cognition, brain function and/or stress, anxiety and depression. A total of *n* = 25 eligible research articles from a possible 3853 published between October 2018 and August 2021 were identified and included. Two study design methods for synthesis were identified: cross-sectional or pre/post intervention. Few cross-sectional design studies that linked microbiota to cognition, brain activity/structure or mental wellbeing endpoints existed (*n* = 6); however, correlations between microbiota diversity and composition and areas of the brain related to cognitive functions (memory and visual processing) were observed. Intervention studies targeting the gut microbiota to improve cognition, brain structure/function or emotional well-being (*n* = 19) generally resulted in improved brain activity and/or cognition (6/8), and improvements in depression and anxiety scores (5/8). Despite inherit limitations in studies reviewed, available evidence suggests that gut microbiota is linked to brain connectivity and cognitive performance and that modulation of gut microbiota could be a promising strategy for enhancing cognition and emotional well-being in stressed and non-stressed situations.

## 1. Introduction

Over the past 10 years, the gut-brain-axis (GBA) and its relationship to health and disease of the brain has received considerable attention [1,2,3]. Segments of both the central and enteric nervous system form an integral part of the GBA, allowing cerebral regions of higher function (i.e., hippocampus and amygdala) to be connected with peripheral intestinal function [2,4]. Recently, the term microbiota-gut-brain-axis (mGBA) has been recommended to better reflect this relationship with reviews by Cryan et al. [5] and Margolis et al. [6] providing detailed overview of the axis and its development. In brief, the mGBA provides an established bidirectional communication pathway between brain and microbiota to monitor and integrate functions of the gut, but also connect cognitive and emotional centres of the brain with peripheral intestinal functions [7]. Recent advances in clinical and experimental research have described the importance of microbiota bacteria in influencing these interactions, highlighting the role of their functional secretory capacity of neurotransmitters and neuromodulators, amino acids and other compounds including short-chain fatty acids (SCFAs) and folate [8,9]. These bacteria can interact locally through direct stimulation of intestinal and/or immune cells to release pro- or anti- inflammatory cytokines and subsequently alter immune function, [10] and intestinal permeability [11], but also distantly with the central nervous system which can lead to functional/connectivity change [12,13]. Microbiota bacteria have been implicated in the development and stability of lacteals and thus play a role in lymphatic function and whole body homeostasis [14,15]. 

While the composition of microbiota bacteria is relatively stable throughout life, it is sensitive to environmental change and can rapidly respond to both internal and external factors such as stress (sleep quality, chemicals, physical and psychological stressors), nutrition (healthy vs. unhealthy eating patterns) and medication [16]. These stressors can alter the gut bacterial balance, leading to low diversity and uneven distribution of bacterial species usually in the direction of non-commensal bacterial communities [17]. This is known as ‘dysbiosis’ and while there is no current consensus on a composition that defines a ‘healthy gut’, a bacterial composition that displays diversity and evenness is a strong candidate [18]. Gut dysbiosis is characterized by altered host immune function, energy metabolism and intestinal epithelial cell damage resulting in increased intestinal and systemic inflammation [19]. This has been linked to many disease states including inflammatory bowel disease, coeliac, non-intestinal auto-immune diseases, obesity and cancer [20,21,22], and, more recently, psychological conditions such as anxiety [23] and depression [24]. Given recent suggestions that the gut microbiota could be an underappreciated mediator of stress responses and associated changes in cognition, mood and well-being [16], considerable attention has been made at identifying therapeutic options for potentially negating the negative effects of such stressors.

The use of probiotics that target and modulate the microbiome have been the focus of recent years with preclinical studies demonstrating its influence on brain development, function and behaviour [25,26,27]. In 2013, the term ‘psychobiotics’ was coined to describe any exogenous influence (i.e., probiotics, prebiotics, dietary fibre) that confers mental health benefits to host that is bacterially mediated [28]. While the mechanisms of action has not been completely elucidated, their benefits maybe attributed directly to production of neurotransmitters and neuromodulators, anti-inflammatory cytokines and/or a reduction in gut barrier dysfunction [29,30]. In 2020, Tooley et al. [31] undertook a narrative review to examine the relationship between the human gut microbiota, including modulators of the microbiota on cognition and/or brain function, which included studies published from 2010–2018. Despite limited studies at the time, relationships between microbiota diversity and enhanced cognitive flexibility and executive function with several bacterial phyla were identified [31]. Further, the majority of prebiotics, probiotics and synbiotics supplementation studies led to positive effects in improving brain function and cognitive processes, such as memory, verbal learning, emotional reactivity, and attentional vigilance [31]. 

Within the past 4 years, there has been an exponential increase in papers published on this topic. Moreover, the emerging role of the oral microbiota in neurological function has gained recent attention. The current systematic review will extend on the narrative review findings of Tooley et al. [31], but undertake a systemic approach and focus on new studies published since October 2018. 

## 2. Materials and Methods

A stepwise approach was used to identify articles from databases following the guidance of Preferred Reporting Items for Systematic Reviews and Meta-Analyses Protocols (PRISMA-P) [32]. This review was initially submitted for registration with the International Prospective Register of Systematic Reviews (PROSPERO) on the 28 September 2020, ID CRD42020206173. Subsequent additions to the review were made and the PROSPERO registration was updated to reflect the new dates of review on the 30 August 2021. Due to the COVID-19 pandemic, PROSPERO registrations at this time were not checked for eligibility, and the registration record was automatically published exactly as submitted. 

### 2.1. Identification and Screening of Relevant Studies

#### 2.1.1. Search Criteria

A wide electronic database search was conducted via SearchLight, a multidisciplinary search platform which accessed literature from Scopus, Medline, Complementary Index, OpenAIRE, Science Citation Index, Supplemental Index, Springer Nature eBooks, arXiv, Social Sciences Citation Index, Knovel and IEEE Xplore Digital Library for the purposes of this review. The same keyword search terms as per previous publication [31] was used on two separate occasions: 1st search -15 September 2020 for items listed from the 30 October 2018–September 2020; 2nd search—October 2020–31 August 2021. Search rules for a positive result included: required to have an item identified from “list 1” and “list 2” (See Table 1). Noting considerable overlap of research on cognition with psychological conditions, additional terms were employed to ensure the widest capture of relevant data.

The search also designated that the terms ‘AD, Alzheimer’s, Parkinson’s, hepatic encephalitis, elderly, autism’ were excluded from results.

Given we were building on the previous work of one of the authors (K.L.T), preliminary searches were commenced and piloting of the study selection process prior to undertaking this review. K.L.T previously published a scoping review on records obtained using the same search terms, published prior to the 30 October 2018 [31]. As described by Moher et al. [32], missed items from the following sources were added manually: reference lists from known published literature reviews, known existing networks, organisations, conferences and authors. These items are captured in Figure 1 as “other sources”.

At both search time-points, a total of 2200 and 1653 articles, respectively, were retrieved from the search. Duplicates were removed (as depicted in Figure 1) and remaining items were scrutinized for inclusion by SC by assessment of “title” (1st pass); abstract screening for relevance (2nd pass) and finally full-text review (3rd pass). Three researchers undertook the abstract review independently, such that each record was reviewed by at least 2 authors. 

#### 2.1.2. Data Extraction for Analysis

Endnote X9™ was used for synthesis and collation of data. Following the same format as Tooley et al. [31], papers were divided into (1). correlation (exploratory) and (2). intervention(s). Similarly, studies where an intervention was implemented, the type of intervention(s) (prebiotic, probiotic, synbiotic, postbiotic and/or paraprobiotic) was noted and used as categories for reporting and data comparison and discussion purposes. Additional information related to trial design, participant numbers and demographics, dose/frequency, and assessments in relation to the two overarching study designs themes were extracted and included for synthesis and comparison of findings. In the event that missing data or incorrect information was identified, the authors of the article were contacted to provide such information. 

#### 2.1.3. Study Risk of Bias and Quality Appraisal Assessment

Information pertaining to blinding, randomisation and appropriate control/placebo groups were captured in data synthesis. Omission of required detail was indicative of lower-quality papers or higher risk of bias. The Revised Cochrane Risk of Bias (RoB 2.0) ‘tool for randomised trials’ was used for assessment of parallel-group studies, and the Rob 2.0 ‘individually randomised, cross-over trials’ tool was used to assess trials that utilised a cross-over design. The Cochrane risk-of-bias tool is most commonly used tool for randomised trials [33]. The JBI Critical Appraisal Checklist for analytical cross-sectional studies was used to assess bias in included correlational studies. The JBI tool is appropriate for use in evidence-based practice and biomedical sciences [34]. Parallel RCT trials were also assessed for conflict of interest bias (i.e., funding, CEO of supplement companies listed as co-authors, etc.), using domain 4 ‘Risk of bias in measurement of the outcome’.

## 3. Results

### 3.1. General Data Extraction

As depicted in Figure 1, a total of 3855 items were identified using the applied search terms and rules. 3446 were removed as duplicates; 232 were deemed “not appropriate” at title screening; 116 were excluded via abstract screening (various reasons recorded in Figure 1). A remaining *n* = 61 research items were included for full-text review. A further *n* = 36 were excluded due to outside age range (×11), medical condition (×12), not relevant (×7), not peer reviewed, full article could not be located (× 2) and other (× 3) (A list of excluded papers and reasons for exclusion can be found in Appendix A). This left *n* = 25 included papers for full review and synthesis. Please note that one study [35] includes study participants that did not meet the inclusion criteria and thus, only data from the individuals that met our inclusions criteria were extracted and included for synthesis and comparison. 

### 3.2. Cross-Sectional Study Designs

Six eligible studies examined the relationship between the microbiota and/or metabolite by-products to aspects of cognition, brain structures/function (as measured by functional Magnetic Resonance Imaging (fMRI)) and markers of stress and well-being (both biological and perceived/self-reported) were included [36,37,38,39,40,41]. Further study details are provided in Table 2. 

Two studies explored the relationship between brain activity (resting state functional connectivity (via MRI)) and bacterial microbiota diversity and composition in healthy adults [36,39]. Resting state functional connectivity between the insula and several regions of the brain linked to important cognitive functions (i.e., multitasking control, episodic memory retrieval, mentalizing evaluation and integration of interoceptive information) was associated with bacterial microbiota diversity and structure [36]. When smoking status was factored in the analysis, two clusters of bacteria genera, *Prevotella* and *Bacteroides* were revealed to be associated with insular connectivity, with higher *Bacteroides* and *Prevotella* associated with lower and higher connectivity, respectively [36]. These exploratory findings suggest possible tobacco-related alterations in bacteria genera. In a similar population demographic, differences in resting state functional activity between adult smokers and non-smokers were explored [39], however this relationship was investigated in the oral-derived microbiota. Relative abundance of *Treponema* (class *Spirochaetes*), and *TG5* (class *Synergistia*) were positively associated with the functional network connectivity, while *Neisseria* (class *Betaproteobacteria*) demonstrated an opposite relationship. However, similar to the Curtis et al. study, smoking status influenced these microbiota-brain connectivity associations [39]. 

Two studies examined the association between self-reported feelings of emotional well-being and gut microbiota profiles (i.e., composition and diversity) [38,41]. The first study was among healthy Korean adults, which demonstrated an association between species diversity (Shannon Index) and “feeling good” (Positive and Negative Affect Schedule (PANAS)), particularly in *Prevotella*-predominant individuals [38]. Of interest was the identification of a novel genus in *Lachnospiraceae* family and known butyrate producer—PAC001043_g, which was significantly associated with both higher positive and lower negative affect scores [38]. In the second study, Taylor et al. [41] found relative abundance levels of bacteria to be related to subjective mood states in healthy American adults without diagnosed mood disorder, but such relationships appeared to differ by sex and influenced by dietary fibre intake. In females, higher relative abundance of *Paraprevotella,* and *Dialister* was associated with higher depression, anxiety and/or stress scores, whereas *Proteobacteria* was associated with lower depression, anxiety and/or stress scores (via Depression, Anxiety and Stress Scale (DASS-42)). In males, higher relative abundance of *Rikenellaceae, Dorea,* and *Blautia* were associated with lower depression or anxiety scores [41]. 

A pilot study conducted in non-obese and obese adults found associations between higher faecal acetate levels and plasma glutamate/acetate ratio to slower Trail Making Test (TMT)-B scores (assessing executive function/cognitive flexibility [42]) and between relative abundance of *Streptococaceae*, lower faecal glutamate levels and slower TMT-A scores (assessing rote memory) [40]. Conversely, faster TMT-A (trend) and TMT-B scores were associated with relative abundance of *Coriobacteriaceae*, higher plasma and faecal glutamate levels. Additionally, associations between higher faecal glutamate levels, glutamate/glutamine ratio and faster TMT-A and TMT-B scores, as well as associations between *Corynebacteriaceae* and *Burkholderiaceae*, faecal glutamate levels, glutamate/glutamine ratio and faster TMT-A scores were also noted [40].

Finally, Langgartner et al. [37] examined the relationship between composition of the oral microbiome as a result of upbringing (urban vs. rural) and immune responses to acute psychological exposure (Trier Social Stress Test (TSST)). Participants that were raised with absolutely no animal contact until the 15th birthday “urban” displayed a significantly different microbial β-diversity along with a more pronounced immune activation (greater increase in peripheral blood mononuclear cell counts (PBMC)) following the TSST test compared to those urban participants reporting daily or occasional animal contact [37].

### 3.3. Intervention Study Designs

Nineteen eligible studies examined the influence of microbiota modulators on brain structures/function and/or aspects of cognition and/or markers of stress and emotional well-being (both biological and perceived/self-reported) were included [35,43,44,45,46,47,48,49,50,51,52,53,54,55,56,57,58,59,60]. Microbiota modulators ranged from a single-species, multi-species probiotic, prebiotic, paraprobiotic and/or postbiotic. Results have been addressed according to the types of intervention implemented and further details are provided in Table 3.

#### 3.3.1. Probiotics

Eleven studies used a probiotic intervention [43,45,46,48,49,51,55,57,58,59,60]. Seven studies employed a single-species probiotic intervention [45,46,48,49,55,59,60], where two different single-species of bacteria (from either the *Bifidobacterium* or *Lactobacillus* genus) were most commonly assessed. Four studies implemented multi-species probiotic interventions and included bacteria from *Bifidobacteria, Lactobacilli, Lactococcus* or *Bacillus* genera [43,51,57,58]. Three studies used inactivated probiotic strains paraprobiotic [47,50,53] and one study used paraprobiotic and postbiotic (*lactobacillus* fermented *Saccharina japonica* (FSJ) extract) [52]. A postbiotic includes concentrated fraction of fermented / secreted products from probiotic bacteria [61]. From a clinical safety perspective, no significant medical side effects were noted.

##### Single-Species Probiotic Intervention

In moderately stressed young adults, 12 weeks of daily consumption of *Lactobacillus plantarum* DR7 (1 × 10^9^ CFU/day) significantly improved the speed needed for social emotional cognition, verbal learning and memory while reducing errors (non-significantly) for associate learning compared to the placebo as assessed by CogState [46]. These effects were possibly due to upregulated serotonin pathways and dopamine pathways stabilization along the GBA. In addition, *L. plantarum* DR7 supplementation reduced symptoms of stress, anxiety, and total psychological scores as well as lowering plasma cortisol and pro-inflammatory cytokine levels (interferon-γ and transforming growth factor-α) and increasing anti-inflammatory cytokines levels, such as IL-10 [46]. Interestingly, older adults (>30 years old) appeared to demonstrate better improvements in basic attention, emotional cognition, and associate learning compared to the placebo and young adults. 

In a follow up publication reporting on secondary outcomes [48], DR7 supplementation appeared to prevent shifts in microbial community compositional (as a result of stress) as indicated by greater α-diversity and differences in β-diversity compared to placebo at the end of 12 weeks. Reduced abundance of *Bacteroidetes, Bacteroidia, and Bacteroidales* in the placebo group was correlated with higher gene expression of a dopamine-pathway enzyme, dopamine beta-hydroxylase (DBH), an indicator of higher stress. Maintained levels of *Bacteroidia* and *Bacteroidales* in the DR7 group were correlated with higher levels of markers of the serotonin pathway [48]. Increased abundance of Firmicutes phylum (i.e., *Blautia, Clostridia, Romboutsia*) were correlated with higher levels of DBH in the placebo group, with *Blautia* correlated with gene expression markers of serotonin pathway. Conversely, increased abundance of *Romboutsia* was correlated with lower levels of an enzyme involved in conversion of tryptophan to serotonin in the brain (tryptophan hydroxylase-2 (TPH2)). Increased abundance of *Negativicutes* and *Deltaproteobacteria* following DR7 supplementation was associated with higher levels of Tryptophan hydroxylase 2 (TPH2), but lower levels of DBH (*Deltaproteobacteria* only) [48]. 

Similarly, Ma et al. [60] performed a secondary analysis of their previously published research [63], which identified a potential link between probiotic-induced gut microbiota modulation and stress/anxiety alleviation in stressed adults. Placebo-receivers showed a significantly larger difference in the Aitchison distance (in microbial structure and composition between two microbiota communities) and Shannon Index between weeks 0 and 12 compared to individuals receiving *L. plantarum* P-8 (2 × 10^10^ CFU/day). More specifically, relative abundance of *Bifidobacterium adolescentis*, *Bifidobacterium longum*, and *Fecalibacterium prausnitzii* were significantly increased, while the abundances of *Roseburia faecis* and *Fusicatenibacter saccharivorans* were significantly decreased in the probiotic group. *B. adolescentis, B. longum,* and *F. prausnitzii* were negatively correlated with stress/anxiety symptom scores, while *R. faecis* was positively correlated with such scores [60]. Average predicted abundances of neuro-related metabolites (cholate, arachidonic acid, creatine, threosphingosine, erythronic acid, and C18:0 sphingomyelin) were significantly higher in the probiotic group compared to placebo [60]. 

In a series of studies, the effects of daily probiotic *Bifidobacterium longum* was assessed on brain activity [59], cognition [45,49], immunity and inflammation [45] and measures of stress and mood [49,55,59], where it’s implementation as an intervention produced mixed results.

Carbuhn and colleagues [45] showed that *Bifidobacterium longum* 35624 supplementation (1 × 10^9^ CFU/day) during offseason training in collegiate female athletes reduced serum IL-1ra and salivary IgA (trend) levels at mid-training (week 3) compared to placebo group. This aligned with significantly higher values of conflicts/pressure in week 3 and significantly lower values in social stress at end of week 4 assessed by cognitive RESTQ52-sport. No statistical differences between groups in serum markers of GI integrity during training were noted. Following an intense training regime during the final 2 weeks of training, significantly higher values in personal accomplishment scale (week five) and self-regulation scale (weeks five and six) under the ‘Sport Recovery’ category were observed in the *B. longum* 35624 group compared to placebo group potentially indicating an ability to mentally push and motivate oneself more [45].

Two studies used a similar dosage of *B. longum* (1 × 10^9^ CFU/day) 1714 for 4 (parallel design) [59] and 8 weeks (cross-over design, 4 weeks washout) [49] in healthy adults in response to stressful situations (social stress game or university exam period, respectively). In resting state, Wang et al. [59] showed *B. longum* 1714™ feeding significantly altered neural activities linked to cognitive control—increased theta band (6 Hz) power and reduced beta-2 band (26 Hz) power. In response to social stress (“Cyberball game”), *B. longum* 1714™ increased source power in theta band (6 Hz) and alpha band (11 Hz) in various brain regions. Change in resting brain activity by *B. longum* 1714™ was associated with the SF36 scale “Energy/Vitality” [59]. In the second study, Moloney et al. [49] found *B. longum* 1714 supplementation was unable to reduce self-reported markers of stress and anxiety and cortisol awakening levels, or alter microbiome species diversity and cognitive performance in young healthy university students undertaking exams [49]. Despite the lack of effect, moderate improvement in sleep quality was noted in individuals receiving *B. longum* 1714 during the exam period, when controlling for baseline scores [49]. 

Finally, a study assessed the effects of *B. longum* (~4 × 10^10^ CFUs/day) supplementation for 7 days on self-perceived stress, depression, and anxiety levels in young university students. Not surprisingly, this short supplementation period failed to have any effect when compared to placebo [55]. 

##### Multi-Species Probiotic Intervention

One study published additional findings from their 2018 study [62] examining the effects of a probiotic stick containing nine different probiotic strains at a dose of 7.5 × 10^6^/day for four weeks in young healthy adults. Results from the 2019 study [43] showed no changes in structural connectivity, but significant changes in functional connectivity following probiotic supplementation compared to placebo and control [43]. Several brain networks were impacted, including the (large-scale) salience network. The salience network, along with its interconnected networks including the insula, is important for task switching, modulating behaviour and more efficient attentional control [43].

In young women with polycystic ovary syndrome, 12 weeks of co-supplementation of Vitamin D (every 2 weeks) and probiotic (daily) containing four viable and freeze-dried strains (2 × 10^9^ CFU/g each) significantly reduced beck depression inventory (BDI), general health questionnaire-28 (GHQ-28) and DASS scores, but not indicators of sleep quality (PSQI) [51]. These changes were supported by significant reductions in systemic markers of inflammation and oxidative damage (high sensitivity C-reactive protein (hs-CRP) and malondialdehyde) and significant elevations in antioxidant levels/capacity levels (glutathione and total antioxidant capacity) [51]. 

Finally, two studies investigated the efficacy of a multi-strain probiotic product on healthy individuals with self-reported anxiety [57] or facing university examinations (stress) [58]. The first study used a cross-over design to assess the effectiveness of 6 strains with 2 × 10^9^ CFU/10 mL (Lactoflorene^®^ Plus—See Table 2 for breakdown) for 45 days, separated by a wash-out period of 25 days [57]. Despite participants meeting the entry score requirement for self-reported anxiety via State-Trait Anxiety Inventory (STAI) scale (≥35 for men and ≥40 for women), this was not confirmed by serology at baseline. Lower natural killer (NK) cell activity in the probiotic group compared to placebo group was observed. However, interpret results with caution given significant differences in markers (including cortisol) indicate the 25 day wash out period was not sufficient. The probiotic group demonstrated increased anti-inflammatory bacteria (*Faecalibacterium* spp.) and reduced pro–inflammatory populations (eg. *Dialister* spp.), reductions in abdominal pain, increased IL-10 and faecal IgA levels. Conversely, the placebo group showed an increase in potentially detrimental or pro-inflammatory bacteria [57]. The second study used a parallel study design and found that daily supplementation of multi-strain probiotic that contained similar strains to the Soldi et al. study (1 × 10^10^ CFU/day) for 28 days was effective at significantly reducing moderate stress (scored on PSS-10 and DASS), and high anxiety (STAI) levels and cortisol levels in university students facing examination [58]. 

##### Paraprobiotic

The effects of 2 weeks supplementation of an inactivated probiotic strain of *Bacillus coagulans* (BC) (1 × 10^9^ CFU/day) was examined in male soldiers participating in a combat training course [47] where only an *n* = 8/group. Whilst no significant differences were observed between the two treatment groups, and intervention period would be deemed “short”, trend improvements in power jump, anti-inflammatory cytokine IL-10, TNFα and IFγ concentrations were evident compared to placebo [47]. 

Two studies investigated the effects of heat-inactivated *Lactobacillus gasseri* strain (CP2305), an anaerobic Gram-positive bacterium, on elite athletes [53] and young adults preparing for medical practitioner exams [50], an extension on previously published works [64,65]. 

In elite athletes, daily intake of the paraprobiotic CP2305 (1 × 10^10^ bacterial cells/day) in beverage form for 12 weeks mildly improved recovery from mental fatigue (Chalder Fatigue Scales), but significantly relieved anxiety and depressive mood (indicated by HADS and STAI-strait scores) during intensive training [53]. Additionally, CP2305 supplementation significantly reduced levels of the salivary stress marker, chromogranin (Cg) A, and lymphocyte and eosinophil counts compared to placebo controls. Finally, richness and evenness of the gut microbial ecosystem was improved following CP2305 supplementation, as well as significant increases in *Faecalibacterium* composition and prevention of *Bifidobacterium* reduction (which was observed in the placebo group) [53].

In young adults preparing for the national medical examination, Nishida et al. [50] demonstrated that CP2305 intake (1 × 10^10^/day) significantly reduced STAI-trait anxiety scores and ameliorated anxiety and depressive mood (Hospital Anxiety and Depression Scale (HADS) questionnaire) relative to placebo. CP2305 supplement also significantly improved sleep quality (PSQI) and lowered depressions scores (GHQ-28) which corresponded with reduced salivary CgA levels. No significant difference in salivary cortisol levels between the two groups were noted. CP2305 significantly mitigated the reduction in *Bifidobacterium* (similar to their previous study Sawada study in runners) and prevented the elevation of *Streptococcus*. Among the SCFAs measures, only n-valeric acid concentrations were significantly increased in the CP2305 group compared to placebo [50]. 

Finally, in young healthy adults, 4-weeks of daily consumption of a synbiotic that contained *Saccharina japonica* (FSJ) extract (postbiotic) fermented by paraprobiotic *Lactobacillus brevis* (*lactobacillus* FSJ—500 mg) improved some aspects of cognitive function as indicted by increased percentage of correct answers, concentration for space perception for memory ability and space perception ability [52]. However, this was only within group changes, with no significant differences between groups. Despite reduced serum amyloid-β concentrations (32%) and increased concentrations of antioxidants (20.0%) in the intervention group, these were not statistically significant between the two groups [52]. 

#### 3.3.2. Prebiotic

Four studies used a prebiotic intervention [35,44,54,56]. Two studies examined galacto-oligosaccharides [35,54], one study used oligofructose-enriched inulin [56] and one study used a polydextrose fibre (PDX) [44]. Three studies employed a cross-over design, with the other study assessing acute effects (1 day—indicative of a non-microbiome influence) [56]. Washout ranged from a minimum of 3 weeks [54], with an average of 4–6 weeks. 

Two studies examined the effects of galacto-oligosaccharides, with Schaafsma et al. [54] focusing on sleep disturbances, while Wilms et al. [35] focused on microbiota composition and systemic markers of inflammation and immune cell modulation. Results from the Schaafsma et al. [54] study showed that 3 weeks (with 3 week wash-out) of a dairy product (DP) that included a prebiotic (5.2 g Galacto-oligosaccharides (BiotisTM GOS), 70% pure GOS) 1 h before bed-time did not significantly affect absolute PSQI scores (and sub-scores) in adults with sleep disturbances (PSQI > 9). However, analysis of intervention period 2 only revealed DP resulted in lower PSQI scores compared to placebo at day 14. When comparing within subject changes at each time point with baseline, only DP showed a significant decrease in PSQI scores at every time point. Gut microbiota profiles were not significantly different between groups at day 21. Nevertheless, within the DP group, there was a significant difference between baseline and day 21 (variation explained 3.0%) in which day 21 was associated with higher relative abundance of *Bifidobacterium*. Low relative abundance of *Bifidobacterium* at baseline was a predictor for sleep improvement, irrespective of treatment [54]. 

Wilms et al. [35] used a higher dosage (15.0 g/day GOS) and longer supplementation period (4 weeks, with 4–6 weeks washout), but found no benefits on stimulated plasma cytokine production and serum CRP concentrations in healthy adults (indicative of immune cell activity and inflammatory levels). Similar to the Schaafsma et al. study, relative abundance of *Bifidobacterium* increased following GOS supplementation (at one and four weeks), which was confirmed by faecal qPCR. However, microbial diversity decreased significantly after 4 weeks of GOS supplementation [35]. 

One study employed a very acute ingestion protocol (1 day) of 13 g of oligofructose-enriched inulin (Orafti^®^Synergy1) on self-reported digestive symptoms, mood and cognitive performance [56]. Ratings of alertness and hedonic tone (i.e., feelings of pleasantness or happiness) were lower following inulin consumption versus placebo. Cognitive performance tests revealed poorer recall accuracy during episodic memory tasks and slower semantic processing speed in the inulin group compared to placebo [56]. Given observed changes occurred quickly following ingestion of a prebiotic, it is likely due to poor study design, rather than changes to the gut microbiome. 

Finally, Berding et al. [44] employed a cross-over study design (4 week intervention and washout period) to examine the effects of 12.5 g of Litesse^®^Ultra (> 90% PDX polymer) on cognitive performance, mood, acute stress responses, microbiota composition, and inflammatory markers. A mild improvement in cognitive flexibility and sustained attention was observed after PDX supplementation. Acute stress (cold pressor test) did not change circulating levels of classical monocytes (CD14+, CD16-), but did increase the expression of the adhesion receptor CD62L on classical monocytes in the placebo group, with no change in the PDX group. No changes in α- and β-diversity, but there was a significant increase to the genus *Ruminiclostridium* 5 after PDX supplementation [44]. 

### 3.4. Risk of Bias and Quality Appraisal Assessment 

Quality appraisal for cross-sectional studies using JBI Critical Appraisal for Analytical Cross-sectional Studies are displayed in Figure 2A. The risk of bias for parallel intervention and cross-over study design using RoB 2 are displayed in Figure 2B,C. Of the cross-sectional studies, the majority of studies addressed the criteria and thus were rated quite high. Only 2/6 studies did not address the first 2 criteria. For the parallel and cross-over study designs, 5 studies did not provide details of the sequence generation process [43,53,56,59,60], with another 2 providing limited detail [47,50]. Three studies did not provide the method of allocation concealment [47,56,60], with another 1 describing limited detail [50], which lead to an unclear risk of selection bias. Risk of blinding the participants was low, but the risk of blinding to study personnel was high, with 9/20 not indicating or providing limited information [43,45,47,49,50,53,56,59,60]. The risk of outcome assessment was mostly low. However, one study [56] was an unblinded controlled trial, leading to a high risk of performance and detection bias. Reporting bias was high as disclosed funding from supplement/product company was considered a potential bias. Ten/nineteen studies were funded by the supplement/product company [35,43,45,46,47,48,50,53,54,56], with another 7 studies listing co-authors that worked or have been funded before by said supplement/product company [46,50,53,54,57,58,59]. Only 11/19 studies registered their trial with their respective clinical trials registry [35,46,48,50,51,53,54,57,58,59,60]. 

## 4. Discussion

The purpose of this systematic review was to determine the following in healthy individuals: (1) are specific microbiota bacteria or microbiota ‘signatures’ (multiple bacteria genera) linked to desirable brain functionality, cognition and emotional well-being; (2) does interventions that target the gut microbiota translate to enhanced cognitive performance and/or improved levels of perceived stress, anxiety and/or depression; and (3) if effective, identify potential mechanisms of action within changes in microbiota diversity and composition, metabolites, systemic inflammation and/or immune modulation. Furthermore, our review endeavoured to focus on generally healthy individuals (i.e., no significant clinical diagnosis), predominantly young adults, in an attempt to reduce the complexity of our analysis and improve the applicability of our findings to individuals that work in stressful and challenging environments, such as first responders, emergency response providers and warfighters. 

### 4.1. Summary of Main Results 

The 6 cross-sectional studies that explored correlations between microbiota cognition/brain function, including depression, anxiety and stress were generally well designed. Significant relationships between gut and oral microbiota diversity and resting state functional connectivity (via magnetic resonance imaging) were identified between the insula and several regions of the brain. Specific bacteria genus (*Bacteroides* and *Prevotello*) in both oral and gut microbiota were related to connectivity between parts of the brain such as middle insula, frontal pole, left angular gyrus and right fusiform/lingual. The insula region and left angular gyrus integrate and process information, including recognition and attention, while the frontal pole and fusiform/lingual gyrus may underlie important cognitive functions, including control over multitasking, episodic memory retrieval, and visual or language information processing [36,66]. Lower activation of such regions could occur during stressful situations. Some associations were moderated by smoking status suggesting potential pathways from smoking (i.e., inflammatory pathways) could be influencing the interactions between microbiome and neurological activities (i.e., recognition and visual processing) [2]. Significantly tighter connections between “feeling good” (emotional wellbeing) and the well-being of the microbiota community (i.e., greater diversity), particularly in the *Prevotella*-dominant condition were evident. Higher relative abundance levels of bacterial SCFAs producers (i.e., butyrate) were associated with low negative affects scores (*Agathobaculum* and PAC001043_g), while pro-inflammatory genus (*Collinsella*) were associated with low positive affect scores. Higher relative abundance of *Peptostreptococcaeae* was significantly associated with higher levels of anxiety, whereas higher abundance of bacteria genera *Anaerostipes, Porphyromonadaceae* and *Parabacteroides* were associated with lower levels of depression or stress. The relationship between microbiota (oral and gut) and brain functionally, mood and immune modulation in healthy adults may differ by sex and influenced by dietary fibre intake and an individuals’ upbringing (no animal contact at all compared to at least occasional animal contact until the age of 15).

The intervention studies that examined their respective intervention product in parallel with placebo/control were generally well designed; however, due to omission of critical design information, specifically sequence generation process and blinding of researchers, classification as ‘high’ risk of bias as determined by JBI evaluations were identified. Moreover, given a large proportion of studies were funded by the company testing its product, we evaluated as ‘high’ risk of bias, despite most authors stating that the funding body had no influence over reported results and the authors themselves had no conflicts of interest. The cross-over design studies were all deemed ‘high’ bias risk given the high chance of carry-over effects and the unlikelihood that true baseline values would be re-established despite a min 28 day wash-out period. Most studies adopted a RDBPC design, and daily intervention ranged from a minimum 7 days to 45 days, though the average was 4 or 12 weeks. One exception was the prebiotic study (adopting the acute effect after four to ten hours)—indicating small intestinal microbiota involvement or nil microbiome influence. 

With regard to the reviewed intervention studies, products shown to have a significant and meaningful effect included prebiotic fibre GOS, probiotic bacteria *Lactobacillus, Lactococcus* and *Bifidobacterium* (when details could be derived from text, specific culture derivations collated and are described in Table 2), and paraprobiotics *Bacillus coagulans* (inactivated) and *Lactobacillus gasseri*. Of the eight studies that investigated brain structure/connectivity and/or cognitive function, six studies reported significant to moderate positive effects in improving brain function and/or cognitive processes/outlook [43,44,45,46,52,59]. One study [56] used a prebiotic inulin and assessed cognition after 4–10 h. Given the major flaws in their study design and reporting, it is difficult to ascertain if the results are a valid indicator of large intestine microbiota influence. Two studies observed marked improvements in brain/neural activity and connectivity [43,59], with another two studies demonstrating enhancement of pathways of the GBA [46,48]. Improvements in memory, social emotion, verbal and associate learning, though the latter was marginal, or space perception reported in two studies [46,52].

Of the eight studies that investigated mood changes (i.e., depression and anxiety), six studies demonstrated improvements in depression and anxiety scores. Three studies did not report benefits [54,55,56], with Siegal et al. [55] providing only 1 week of supplementation (*B. longum*) which would not have been enough time to induce significant microbiota shifts in a young healthy population with already high abundance of *Bifidobacterium*. Schaafsma et al. [54] observed a carry-over effect which potentially masked any true benefits, while the reason for null effects in the Smith. [56], study have already been discussed above. Four studies demonstrated reductions in perceived stress, with one study reporting no effect [55]. Of the six studies that examined inflammatory (i.e., cytokines) or biological stress makers (i.e., cortisol), all but 1 study [35] demonstrated reductions. Limited studies assessed sleep quality, but improvements were shown with paraprobiotic *Lactobacillus gasseri* strain (CP2305), *B. longum* 1714™ and prebiotic GOS. Importantly, all included studies that reported adverse effects from gut microbiota intervention appeared to be nil. Moreover, no detrimental effects on cognition and/or brain function were observed and interventions were generally well tolerated. 

### 4.2. Proposed Mechanisms of Action 

The majority of intervention studies did not directly examine the mechanisms of action of their investigative product, indeed, not all studies analysed microbiota data to confirm changes in diversity and/or composition, and thus it is difficult to accurately determine the mechanism of action of each supplement. Despite this, a few common themes were identified. Firstly, in stressful situations, it is possible with the correct dosing and duration of intervention, that a probiotic (no specific strain) was able to mitigate shifts in bacteria abundance and composition that would typically occur as a result of the stressor and thus translate to improved or at least maintained cognitive performance (if measured). Secondly, ancillary benefits such as reduced inflammation and/or improved sleep quality could also be mediating the improvements in cognition and/or brain neural activity. Finally, while few studies measured metabolites in the faecal/blood, especially SCFAs, change in bacteria that are known SCFA producers were evident in some studies and early results indicate they may play a role. 

### 4.3. Limitations

The studies included in the review were not without limitations. Firstly, there was a high degree of variation in sample size and participant demographic. Sample size ranged from as low as *n* = 30 to 133 for cross-sectional studies and as low as *n* = 7 to 69 per group; low sample sizes of *n* = 7–9 were identified in two intervention studies [45,47]. Secondly, a large proportion of studies did not capture over time (longitudinal—multiple sampling) or at baseline (cross-sectional), other factors known to modulate the microbiota such as sleep hygiene (and circadian rhythms), dietary/nutritional intake, physical activity and other environmental stressors. These factors could be considered covariates when analysing data. These metrics must be considered for collection in future studies. Thirdly, about half of the intervention studies (10/19) did not examine the changes in the microbiota pre/post supplementation but importantly, undertake strain specific recovery measurements to confirm colonisation in the GI system or when colonisation had ceased. While 16S rRNA is informative, it provides limited resolution and detail. Thus implementing shotgun metagenomics sequencing that can provide finer detail on gut microbiota changes at the species level will be integral moving forward in an effort to inform appropriate formulation of gut-microbiota-modulation treatments. Consideration of appropriate “placebo” needs to be better considered, as this review identified the implementation of non-digestible/synthetic sugars which may augment the gut microbiome most likely in a negative direction. Additionally, including assays such as metabolomics (both targeted and untargeted) and inflammatory/immune cell status and/or function will help establish causal relationships and potential pathways of mechanisms of the gut microbiome on cognition in humans. Finally, while crossover design studies can be a powerful approach given its ability to reduce confounding differences within participants between groups, it was evident in some studies that ‘true baseline’ was never achieved prior to second treatment despite the recommended 28 day washout period. Thus, caution should be made when utilising this study design approach in microbiome/intervention studies. 

### 4.4. Recommendations and Future Directions 

Given the limitations that exist in the reviewed studies and risk of biases identified, it is difficult to make formal recommendations regarding which microbiota targeting intervention (single or multi-strain and/or dosage) is best when it comes to enhancing cognition or brain function. Notwithstanding, it is evident that some positive cognitive effect might be achieved when providing probiotic strains from either the *Bifidobacterium* or *Lactobacillus* genus. 

However, there is still much to do into the future to ensure reliability and specificity underpinning the interaction across the mGBA. Specifically to the effect of prolonged stressors or stressful environments and the translator effect on cognition, there remains a need to better understand the following: (1) examine the acute and chronic effects of complex or layered stressors on the gut microbiota in the general populous (e.g., COVID-19 infection effect), elite sportspersons, para-military cohorts and military; (2) assess and compare the efficacy of interventions (known and novel) to ascertain the superior modulator and in what circumstance; (3) since the gut microbiome impacts and influences multiple physiological systems, studies which expand across fields (i.e., general health, gut health, physical performance, immune/inflammation response/function, nutrition or sleep) would be powerful rather than studies being carried out in isolation; (4) implementing higher fidelity metagenomics sequencing of the gut microbiome is a critical assay for both comparative and intervention studies moving forward.

Finally, develop an appropriate methodological design consensus (i.e., adequate dosing, dosing protocol, appropriate placebo, duration of intervention, inclusion of a late time-point to determine performance fade-effect and/or identify cessation of probiotic intervention near/mid/far colonization)—more uniform designs would better allow for ease of comparisons across studies. This would include the discussion of cross-over studies in gut microbiome studies. Whilst cheaper to run and allow for within subject comparison, present challenges since (1) wash-out periods are generally short and prolonged systemic effects are observed; and (2) more importantly, does the gut microbiome return to true baseline levels once modulated [67]. To circumnavigate these issues, RDBPC studies would be deemed more appropriate moving forward

## 5. Conclusions

The findings of this review extend on previous published reports and recent narrative review examining the link between microbiota to brain performance/function in healthy, stressed, anxious, moody or depressed (mood and mental well-being) individuals. Despite inherit limitations in studies reviewed, the available evidence suggests that gut microbiota is linked to brain connectivity and cognitive performance and that manipulation of gut microbiota could be a promising avenue for enhancing cognition and emotional well-being in stress and non-stressed situations within healthy adults. Given a large majority of studies to date have been limited to animal models and have mostly been observational in a clinical setting usually in a disease condition, there are still many unanswered questions. Moreover, well-designed studies in humans linking the gut to brain performance/function in relatively healthy populations that are stressed, anxious, moody or depressed (mood and mental well-being) and/or are exposed to a stressful situation is often underrepresented in publication and thus needed to drive further meaningful progress in this area.

## Figures and Tables

**Figure 1 nutrients-14-04623-f001:**
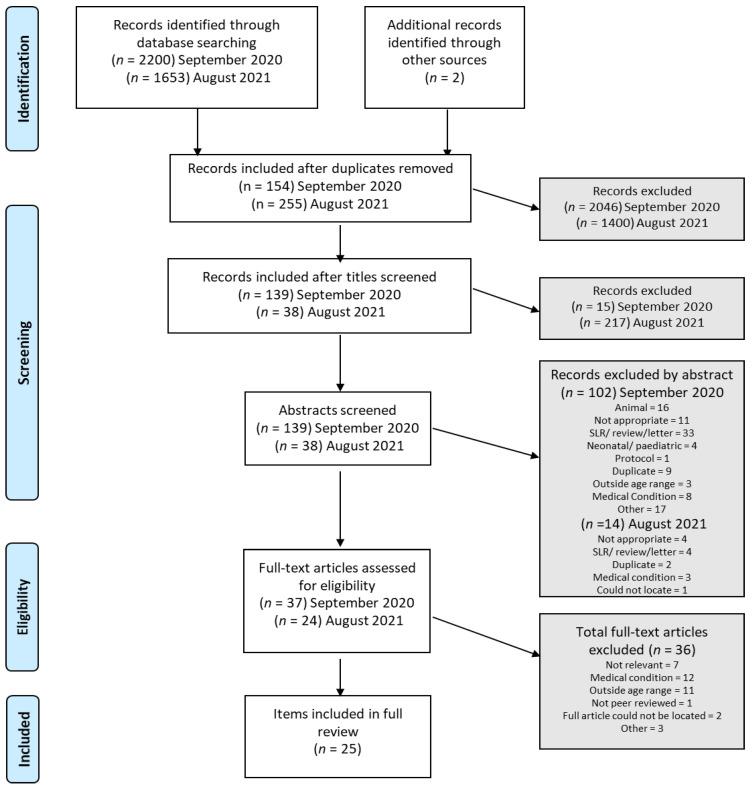
Flow diagram of the study selection process.

**Figure 2 nutrients-14-04623-f002:**
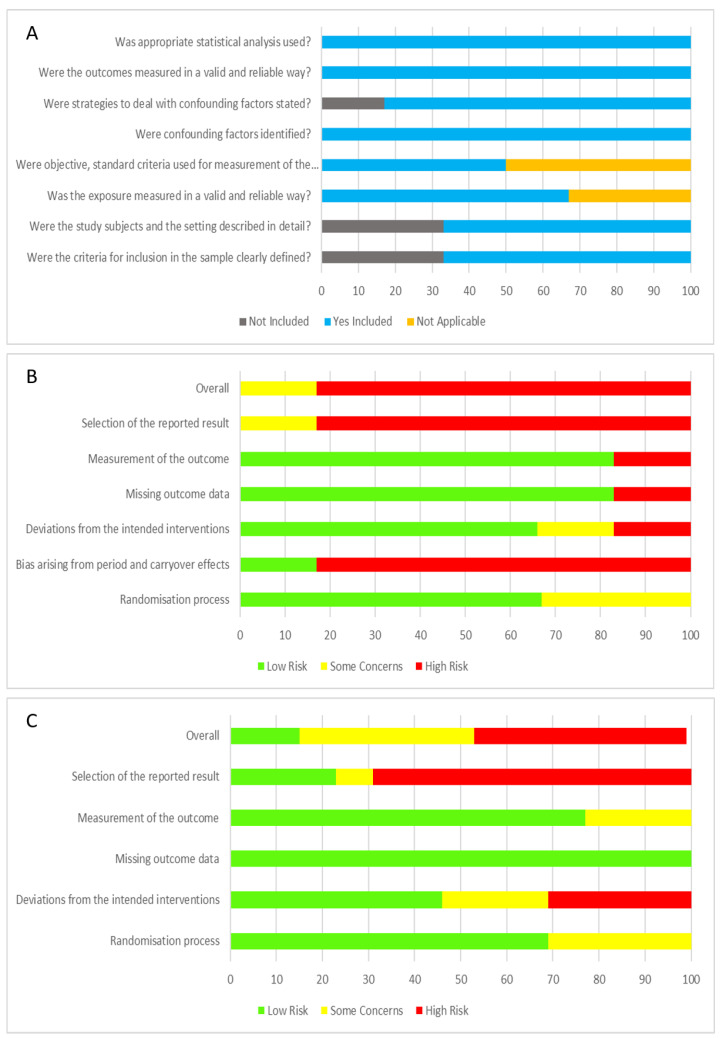
Summary of risk and bias analysis of all studies. (**A**) JBI critical appraisal for analytical cross-sectional studies (*n* = 6); (**B**) Summary of RoB2 risk of bias for cross-over design intervention studies analysis (*n* = 6); (**C**) Summary of Rob2 risk of bias for parallel design intervention studies analysis (*n* = 13).

**Table 1 nutrients-14-04623-t001:** Search list items for literature review.

Search List 1		Search List 2
(“Dietary fib*” OR “inulin” OR “oligo*” OR “Lactobac*” OR “gut permeability” OR “microbio*” OR “Bifidobac*” OR “Streptococ*” OR “prebiotic” OR “probiotic” OR “gut-brain-axis” OR ‘’phytobiotic” OR “paraprobiotic” OR “synbiotic” OR “xenobiotic” OR “psychobiotic” OR “polyphenol”)	AND	(“cognit*” or “memory” or “vigilance” or “decision making” or “attent*” or ”percept*” or “processing speed” or “visuo-spatial” or “executive function” or “task-switching” or “emot*” or “behav*” or “recognition” or “resting-state” or “salience” or “stroop” or “go-no go” or “n-back” or “functional state” or “neuroscience” or “psychobiology” or “stop signal” or “perform*” or “stress” or “cortisol” or ”BDNF” or “serotonin” or “NPY” or “neuropeptide” or “lipopolysaccharide (LPS)” or “lipopolysacc*”)

**Table 2 nutrients-14-04623-t002:** Exploratory human studies describing correlations/interactions between microbiota and cognition, brain structures, function and stress (no intervention).

Author/Year	Participants/Sample (Age M ± SD Years)	Sex (M/F)	Study Design	Assessment	Main Findings—Microbiome Link
Curtis et al., (2019) [36]	*n* = 30; non-smokers *n* = 10 (32 ± 2); eCig users *n* = 10 (30 ± 3); tobacco smokers *n* = 10 (37 ± 3)	28/2	Cross-sectional group comparison	Resting state functional connectivity of the middle insula; faecal microbiota (16S rRNA)	Insular connectivity is associated with microbiome diversity, structure and at least two specific bacteria genera, potentially modulated by tobacco smoking
Langgartner et al., (2020) [37]	*n* = 40; healthy; rural *n* = 20 (25.1 ± 0.8); urban *n* = 20 (24.5 ± 0.8)	40/0	Cross-sectional group comparison with stress test	TSST, saliva (oral) microbiota (16S rRNA), IL-6 and cortisol (plasma) and PMBC	No significant difference in alpha or beta diversity (salivary microbiome). Urban upbringing and neg animal contact had effects on salivary microbiome composition linked to stress-induced immune activation.
Lee et al., (2020) [38]	*n* = 83 (48.9 ± 13.2)	37/46	Correlational; emotional well-being and gut microbiome profiles	Faecal microbiota (16S rRNA), PANAS	Gut microbiome diversity is related to emotional well-being; *Prevotella* was indicative of positive emotional wellbeing
Lin et al., (2019) [39]	*n* = 60; smokers *n* = 30 (37.2 ± 9.6); non-smokers *n* = 30 (37.2 ± 11.8)	smoker 21/8; non-smoker 20/7	Cross-sectional group comparison	Resting state fMRI; metagenome inferred from faecal microbiota (16Sr RNA)	Brain functional component differences linked with smoking related microbiota, indicating smoking induced microbiome dysbiosis and brain functional connectivity alteration
Palomo-Buitrago et al., (2019) [40]	*n* = 35; non-obese *n* = 16 (50.1 ± 10.4); obese *n* = 19 (53.6 ± 5.9)	unknown	Cross-sectional group comparison	Faecal microbiota (shotgun) and plasma and faecal glutamate, glutamine and acetate; TMT-A &TMT-B	Slower TMT-A scores associated with relative abundance of *Streptococaceae* and lower faecal glutamate levels. *Corynebacteriaceae* and *Burkholderiaceae* associated with faecal glutamate levels, glutamate/glutamine ratio and faster TMT-A scores
Taylor et al., (2019) [41]	*n* = 133; 25–45 years (33.4 ± 5.8)	60/73	Exploratory cross-sectional	DASS- 42; faecal microbiota (16S rRNA); dietary intake and diet quality	Bacterial taxa and DASS relationship. Sex associations with bacterial taxa and DASS, inverse relationship between Anxiety scale scores and *Bifidobacterium* (females); inverse relationship with Depression scores and *Lactobacillus* (males).

Acronyms in order of appearance: mean (M); standard deviation (SD); trier social stress test (TSST); peripheral blood mononuclear cell (PBMC); positive and negative affect schedule (PANAS); functional magnetic resonance imaging (fMRI); trail making test (TMT) A or B; depression, anxiety and stress scale-42 items (DASS-42).

**Table 3 nutrients-14-04623-t003:** Human studies listed in alphabetical order describing interventions on the gut microbiota and the effects on cognition, brain structures and function, and stress.

Author/Year	Participants/Sample * (Age M ± SD Years)	Sex (M/F)	Study Design	Treatment/Intervention	Dose/Frequency	Assessment	Main Findings—Microbiome Link
Bagga et al., (2019) [43]	*n* = 45 healthy (26.2 ± 4.8); *n* = 15 no intervention (26.9 ± 5.0); *n* = 15 PLA (27.3 ± 5.8); *n* = 15 PRO (28.3 ± 4.2)	7/8 9/6 7/8 (22/23)	RDBPC, parallel design study	Ecologic^®^825 9 strains: *Lactoba-cillus casei* W56, *L. acidophilus* W22, *L. paracasei* W20, *Bifidobacterium lactis* W51, *L. salivarius* W24, *Lactococcus lactis* W19, *B. lactis* W52, *L. plantarum* W62 and *B. bifidum* W23 (PRO), maize starch and maltodextrins (PLA) or no intervention. See Bagga et al., 2018 [62]	3 g sachet; 7.5 × 10^6^ (PRO) or PLA once daily; 4 weeks. See Bagga et al., 2018 [62]	fMRI resting state and diffusion.	Decrease in functional connectivity in DMN, SN, VIN and MFGN (link to depression and stress disorders) vs. PLA and/or CON.
Berding et al., (2020) [44]	*n* = 18 healthy (26 ± 1.3). Note: *n* = 6 withdrew	0/18	RDBPC, crossover design study	Litesse^®^Ultra (>90%PDX polymer) (PRE) or Maltodextrin (PLA)	12.5 g sachet; PRE or PLA, once daily; 4 weeks, washout 4 weeks before cross over, another intervention 4 weeks	CANTAB tasks MTT, RVP, PAL, SSP, IED, ERT, faecal sample (16S rRNA sequencing), salivary cortisol, cytokines, acute stress response (cold pressor task).	Improved IED (cognitive flexibility) and RVP (sustained attention).
Carbuhn et al., (2018) [45]	*n* = 17 healthy; *n* = 8 PRO, *n* = 9 PLA; age NI. Note: *n* = 3 withdrew(2PRO/1PLA)	0/17	Two-group stratified randomisation, double-blind, placebo-controlled design	*B. longum* 35624 (PRO) or maltodextrin (PLA)	4 mg capsule; 1 × 10^9^ CFU PRO or PLA once daily; 6 weeks	Inflammation (12 cytokines), LPS and LPS Binding Protein, sIgA, cognitive stress-recovery assessment.	No significant effect on exercise performance or immune function. Differences in cognitive outlook between PRO and PLA, especially during intense training phase.
Chong et al., (2019) [46]	*n* = 111 (18–60), PLA *n* = 55 (32.1 ± 11.0); PRO *n* = 56 (31.1 ± 7.8) Note: *n* = 12 withdrew or excluded	NI	RDBPC, parallel design study	*Lactobacillus plantarum* DR7 (PRO) or maltodextrin (PLA)	2 g sachet; 1 × 10^9^ CFU PRO or PLA once daily; 12 weeks	CogState Brief Battery, PSS-10, DASS-42, cortisol, cytokines, plasma neurotransmitters.	Reduced symptoms of stress and anxiety, improved several cognitive and memory functions, reduced levels of plasma cortisol and pro-inflammatory cytokines.
Hoffman et al., (2019) [47]	*n* = 15 soldiers; PAR: *n* = 8 (20.0 ± 0.6); PLA *n* = 7 (20.2 ± 0.6). Note: *n* = 1 withdrew, but included in reported avg. age for PLA	15/0	Double-blind, parallel design study	Inactivated *Bacillus coagulans*, (PAR called Staimune) or PLA (details not specified)	1 × 10^9^ CFU PAR or PLA once daily; 2 weeks	Serum cortisol, testosterone; IL-10., TNFα, IFNγ.	No significant differences between groups. Note: 2 weeks intervention not adequate; not adequately powdered. Trend findings identified.
Liu et al., (2020) [48]	*n* = 111; <30: PLA *n* = 32 (24.9 ± 2.9); PRO *n* = 27 (24.8 ± 2.8). >30: PLA *n* = 23 (41.7 ± 9.5); PRO *n* = 29 (37.0 ± 6.0). Note: *n* = 13 withdrew or lost to follow up (6PRO/7PLA)	NI	RDBPC	*Lactobacilllus plantarum* DR7 (PRO) or maltodextrin (PLA)	2 g sachet; 1 × 10^9^ CFU PRO or PLA once daily; 12 weeks	Faecal microbiota (16S rRNA), gastrointestinal symptoms, stress neurotransmitters.	Changes of gut microbiota along different taxonomic levels; reflective changes in neurotransmitter serotonin and dopamine pathways enzyme gene expression.
Ma et al., (2021) [60]	*n* = 79; PRO *n* = 43; PLA *n* = 36; age for updated dataset NI. Previous study Lew et al., (2019) [63]. Note: *n* = 24 did not provide faecal samples not included in analysis (9PRO/15PLA)	18/61	RDBPC (as per Lew [63])	*L. plantarum* P-8 (PRO) or maltodextrin (PLA)	2 g sachet; 2 × 10^10^ CFU PRO or PLA once daily; 12 weeks	Shotgun metagenomics, metabolomics for gut-brain.	Enhanced diversity of neurotransmitter synthesizing/consuming SGBs and the levels of some predicted microbial neuroactive metabolites (e.g., SCFAs, gamma-aminobutyric acid, arachidonic acid, and sphingomyelin).
Moloney et al., (2021) [49]	*n* = 20 healthy; (20.7 ± 0.28 SEM). Note: *n* = 10 withdrew or excluded	20/0	RDBPC, cross-over design	*B. longum* AH1714 (PRO) or corn starch, magnesium stearate, hypromellose & titanium Dioxide (PRO)	Capsule; 1 × 10^9^ CFU PRO or PLA once daily for 8 weeks, 4 weeks washout before cross over, daily for another 8 weeks	PSQI, PSS-10, CANTAB (visual memory and learning (PAL), sustained attention (RVP), working memory (SSP), emotional recognition (ERT) and social cognition (RMIE)), BDI-II, faecal microbiota (16S rRNA), salivary cortisol.	Stated findings included: no statistical improvement in any cognitive element or the alleviation of stress/anxiety symptomology. *INTERPRET CAUTIOUSLY: Immunological data indicated wash-out period was not sufficient = data not reliable.*
Nishida et al., (2019) [50]	*n* = 60; PAR *n* = 31 (24.9 ± 0.5); PLA *n* = 29 (25.3 ± 0.6)	PAR: 21/10 PLA: 20/9	RDBPC, parallel design	*Lactobacillus gasseri* CP2305 (heat-inactivated) (PAR) or maltose, dextrin, starch, veg oil (PLA)	Per 2 tables; 1 × 10^10^ CFU PAR or PLA, twice daily (2 tablets per day); 24 weeks	STAI, GHQ-28, HADS, PSQI, VAS, salivary cortisol and IgA, CgA, EEG (sleep), faecal SCFA, faecal microbiota (16S rRNA).	CP2305: reduced anxiety; improved sleep quality; reduced GHQ-depression subscores; reduced anxiety and depression (HADS); reduced reactivity physiologically from stress; improved irritability and abdominal discomfort; mitigated changes in microbiota due to stress.
Ostadmohammadi et al., (2019) [51]	*n* = 60; healthy with PCOS; Vit D + PRO *n* = 30 (24.4 ± 4.7); PL *n* = 30 (25.4 ± 5.1)	0/60	RDBPC, parallel design	Vitamin D + *Lactobacillus acidophilus*, *Bifidobacterium bifidum*, *Lactobacillus reuteri* and *Lactobacillus fermentum* (PRO) or corn starch and oil (PLA)	50,0000 IU Vit D every 2 weeks + 8 × 10^9^ CFU (2 x10^9^ CFU/g for each strain) PRO or PLA once daily; 12 weeks	Hormonal profiles, Mental health (BDI, GHQ-28, DASS, PSQI), biomarkers of inflammation and oxidative stress (serum hs-CRP, plasma TAC, GSH and MDA).	Reduced BDI, GHQ and DASS scores compared to placebo. Did not change PSQI score. Reduced testosterone, hirsutism, hs-CRP and MDA levels and increased antioxidant defenses compared to placebo.
Park et al., (2019) [52]	*n* = 39; healthy 18–65 years. SYN *n* = 31 (32.9 ± 17.6); PLA *n* = 32 (31.8 ± 16.3). Note: *n* = 3 from PLA excluded. *n* size used for avg. age and sex ratio are smaller.	SYN: 8/23; PLA: 11/21	RDBPC, parallel design	Fermented *Saccharina japonica* (kelp extract; FSJ postbiotic) by *Lactobacillus brevis* (paraprobiotic) (SYN) or lactose (PL))	Each active capsule contained 500 mg standardized fermented lactobacillus FSJ, 2 x capsules daily; 4 weeks	BDI, K-WAIS, operation-word span task and raven’s test-based quantitative EEG test. Serum amyloid-β, SOD.	Non-significant between groups on cog tests and biochemical measures. FSJ treated group significantly increased the percentage of correct answers and concentration for space perception for memory ability and space perception ability.
Sawada et al., (2019) [53]	*n* = 49, healthy athletes 18–22 years. PAR *n* = 24 (19.8 ± 1.4); PLA *n* = 25 (20.1 ± 1.1).	49/0	RDBPC, parallel design	*Lactobacillus gasseri* CP2305 (heat inactivated) in excipient (PRA) or excipient (PLA). Excipient = isotonic sports drink containing sweetener, acidifier, flavorings, Vit C, and minerals (Na, Ca, K, Mg)	200 mL beverage; 1 × 10^10^ PRA or PLA once daily; 12 weeks	CFS, STAI, HADS, GHQ- 28, PSQI, stress and immune markers (salivary Cg A and immune cells), faecal microbiota (16S rRNA)	CP2305 decreased STAI-state and STAI-trait, improved fatigue, anxiety and depressive mood. Minor changes in bacteria composition
Schaafsma et al., (2021) [54]	*n* = 69; healthy with sleep problems age 30–50 years (M39). Note: *n* = 1 lost to follow-up. *n* = 69 for ITT analysis and *n* = 64 and 47 for PP and mod-PP analysis, respectively.	NI	RDBPC, cross-over design	Dairy-based product (DP) containing protein (Lactium^®^), prebiotic (Galacto-oligosaccharides (BiotisTM GOS), 70% pure GOS, and vitamins and minerals (PRE) or protein (Lactium^®^), vitamins and minerals (PLA)	Sachet; 5.2 g GOS PRE or PLA, once daily; 3 weeks. 3 weeks washout before cross over, another intervention 3 weeks	DASS, PSQI, salivary cortisol; faecal microbiota (16 S rRNA, 1^st^ crossover period only); note: altered endpoint of day 14 was reported on instead of day 21.	Data indicated wash-out was not sufficient and carry-over effects = data contamination. DP reduced salivary cortisol and stimulated *Bifidobacterium* (faecal). *INTERPRET CAUTIOUSLY: contained sucralose which would have confounded gut microbiota; washout not sufficient.*
Siegel & Conklin (2020) [55]	*n* = 79 (19.7); PLA *n* = 39 (19.9 ± 1.1); PRO *n* = 40 (19.4 ± 1.0)	58/21	RDBPC, parallel design, pilot	*B. longum* (PRO) or corn starch (PLA)	400 mg; ~4.0 × 10^10^ CFUs or PLA, twice daily for 7 days	PSS-10; CES-D; STAI.	Non-significant changes in stress, depressive symptoms or anxiety. *INTERPRET CAUTIOUSLY: intervention time-period not sufficient to elicit mental wellbeing matrices assessed.*
Smith (2019) [56]	*n* = 53; 19–54 years (22). No further details provided.	12/39	Placebo controlled cross-over study (blinding NI)	Inulin; Oligofructose-enriched inulin (Orafti^®^Synergy1) (PRE) or maltodextrin (PLA)	13 g (8 g + 5 g, split over 12 h) Acute testing, cross-over assessment next day	Mood (alertness, hedonic tone and anxiety), episodic memory, logical reasoning, semantic processing, SRT, attention lapse, cognitive vigilance	Effect of inulin was (morning): Reduced alertness, reduction in hedonic tone, poorer recall accuracy (episodic memory) and slowed semantic processing. Acute effect, not microflora influence. *INTERPRET CAUTIOUSLY: Not a true prebiotic effect—timeline for effect too acute.*
Soldi et al., (2018) [57]	*n* = 50; 20–35 years. Note: *n* = 6 withdrew/discontinued/antibiotics	NI	RDBPC, cross-over design	Lactoflorene^®^ Plus: *Lactobacillus acidophilus* LA-5^®^, *Bifidobacterium animalis* subsp. *lactis*, BB-12^®^, *Lactobacillus paracasei* subsp. *paracasei*, *L.* CASEI 431^®^, *Bacillus coagulans* BC513, zinc, B vitamins (niacin, B1, B2, B5, B6, B12 and folic acid) (PRO) or zinc, B vitamins (niacin, B1, B2, B5, B6, B12 and folic acid) (PLA)	10 mL liquid; 2 x 10^9^ CFU PRO or PLA, twice daily for 45 days; washout 25 d; crossover to other intervention for 45 days	Salivary stress markers (α-amylase, cortisol, chromogranin A) and immunological parameters (sIgA, NK cell activity, IL-8, IL-10, TNF-α) in faeces, faecal microbiota (16S rRNA), gastrointestinal symptoms.	No direct effect on salivary stress markers or NK cell activity. Reduced abdominal pain and increased faecal IgA and IL-10 levels. Increased anti-inflammatory and reduced pro–inflammatory bacteria with probiotic, reductions in abdominal pain. NK cells indicate wash-out period not adequate. *INTERPRET CAUTIOUSLY: It appears that due to this most results would be skewed.*
Venkataram et al., (2020) [58]	*n* = 74 healthy (21.4 ± 1.5); PRO *n* = 36 (21.2 ± 1.6); PLA *n* = 38 (21.6 ± 1.3). Note: *n* = 6 not allocated a treatment (4PRO/2PLA).	17/63 Note *n* size used for sex ratio based on original *n =* 80.	RDBPC, parallel design	*Bacillus coagulans* Unique IS2, *L. rhamnosus* UBLR58, *B. lactis* UBBLa70, *L. plantarum* UBLP40 (2 billion CFU each); *B. breve* UBBr01, *B. infantis* UBBI01 (1 billion CFU each) capsule with glutamine or microcrystalline cellulose (PLA)	Capsule; 1–2 × 10^10^ CFU PRO + 250 mg glutamine or PLA twice daily for 28 days	PSS-10, DASS, STAI, serum cortisol.	Reduced stress on PSS-10, DASS, and STAI in students facing examination. Early morning, fasting serum cortisol levels decreased compared to placebo.
Wang et al., (2019) [59]	*n* = 40 healthy; 18–50 years; PRO *n* = 20 (31.0 ± 2.3); PLA *n* = 20 (33.0 ± 2.8). Note: *n* = 3 excluded from PRO (antibiotics)	PRO: 7/13; PLA: 7/13	RDBPC, parallel design	*Bifidobacterium longum* 1714™ (Zenflore; PRO) or maltodextrin (PLA)	2 g sachet; 1 × 10^9^ CFU or PLA once daily; 4 weeks	Resting state MEG, MEG during CBG, SF36, social stress induced by CBG, measured by NTS, MQ, and SEP.	*B. longum* 1714 altered resting-state brain activity, and induced change in neural activity correlated with increased energy/vitality. No treatment effect on SF36 or stress.
Wilms et al., (2020) [35]	*n* = 24 heathy adults (38.2 ± 7.8)	Adults (8/16)	RDBPC, cross-over design	Biotis™ galacto-oligosaccharide (GOS) powder (PRE) or maltodextrin (PLA)	7.2 g sachet; 5 g of pure GOS (PRE) or PLA, 3 times daily; washout 4–6 weeks; crossover to other intervention for 4 weeks	Faecal microbiota (16S rRNA) and SCFA, breath volatiles, stimulated cytokines, CRP, MDA, TEAC and uric acid in plasma.	GOS affected microbiota composition, accompanied by increases in bifidobacteria and decreased microbial diversity in healthy adults. Faecal and breath metabolites, immune and oxidative stress markers were not affected by GOS.

* Participants/Sample number reflects final numbers/completions. Acronyms previously not determined: probiotic (PRO); placebo (PLA); synbiotic (SYN); parabiotic (PAR); prebiotic (PRE); control (CON); not indicated (NI); randomised, double-blind, placebo-controlled trial (RDBPC); intention–to-treat (ITT); per-protocol (PP); modified per-protocol (mod-PP); colony forming units (CFU); default mode network (DMN); salience network (SN); visual network (VIN); middle and superior frontal gyrus network (MFGN); symptoms checklist-90 (SCL-90); allgemeine depressions-skala (ADS); leiden index of depression severity (LEIDS); cambridge cognition assessment battery tasks (CANTAB): motor screening test (MTT); rapid visual information processing (RVP); paired associates learning (PAL); spatial span (SSP); intra-extra dimensional set shift (IED); emotion eecognition task (ERT); read the mind in the eyes (RMIE) task; lypopolysaccharide (LPS); high sensitivity C-reactive protein (hs-CRP), total antioxidant activity (TAC), glutathione (GSH) and malondialdehyde (MDA); perceived stress scale-10 inventory (PSS-10); chalder fatigue scales (CFS); spielberger state-trait-anxiety-inventory (STAI); hospital anxiety and depression scale (HADS); 28-item general health questionnaire (GHQ- 28); pittsburgh Sleep Quality Index (PSQI); Standard Error Mean (SEM); salivary immunoglobulin-A (sIgA); Natural Killer T-cell (NK); interleukin (IL-); tumour necrosis factor (TNF); visual analogue scale (VAS); chromogranin A (CgA); encephalogram (EEG); short chain fatty acid (SCFA); polycystic ovary syndrome (PCOS); korean wechsler adult intelligence scale (K-WAIS); superoxide dismutase (SOD); simple reaction time (SRT); magnetoencephalography (MEG); short-form 36 (SF36); cyberball game (CBG); need threat scale (NTS); mood questionnaire (MQ); subjective “exclusion perception” (SEP); centre for epidemiological studies depression scale (CES-D); trolox equivalent antioxidant capacity (TEAC).

## Data Availability

Not applicable.

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
