# Peer review of "Examining the Influence of the Human Gut Microbiota on Cognition and Stress: A Systematic Review of the Literature"

_nutrients, 2022, doi:10.3390/nu14214623_

Round 1
Reviewer 1 Report
In the present manuscript, the Authors summarize the evidence linking the gut–microbiota–brain axis. This review is a well-written summary focusing on relevant experimental work and recent advances in gut microbiota and brain health. The authors should be more accurate in the text deleting a few mistakes and repetitions (i.e., 178 bracket, 502 discussions).
Please try to conclude each chapter with some conclusions. Please try to link all the endings.
An attempt to be more speculative, with some research from a clinical and preclinical point of view where therapeutic strategies can counter or restore brain impairment/stress or anxiety.
Reviewer 2 Report
The paper is very interesting and it discuss a very important topic concerning the role of GUT Microbiota on Cognition and Stress.
The topic is particularly interesting expecially for its impact on healthcare system, quality of life of patients and their proxies.
I coud just suggest sto consider in the introduction the impact of physical activity on physical and psychical outcome (10.1016/j.archger.2020.104109 and 10.1016/j.jamda.2019.01.128)
and the role of polymedication on older adults and mood (10.1017/S1041610217001715 and 10.1007/s40520-018-0893-1).
Reviewer 3 Report
Cooke et al’s manuscript “Examining the Influence of the Human Gut Microbiota on Cognition and Stress: A Systematic Review of the Literature” submitted to “Nutrients” is a comprehensive review of the relationship between human gut microbiota, including modulators of the microbiota on cognition, brain function and stress, anxiety, and depression. The topic represents a particularly up-to-date problem. The review is well-written and the organization is appropriate. The results of the current studies are clearly summarized in the tables. I enjoyed the manuscript and found no big withdraws other than a few minor points and suggestions I indicate below.
1. Table 3: I suggest adding additional rows separating the cited papers by the type of intervention used (probiotic, prebiotic, etc.).
2. In the Results chapter: change “x” to “×” in probiotic dosage (where needed, eg. lines 253, 297).
3. Line 430: Capital letter in „lactobacillus”.
4. Please harmonize captions in Figures 2,3,4 (font size, italic or non-italic).
5. Please remove duplicated the first paragraph of the discussion.
Reviewer 4 Report
The manuscript is interesting and well written. Perhaps in some parts is excessively long and with numerous subsections that could in some cases be unified and in others even eliminated. With respect to the content, the work is well done and despite the limited number of studies that were finally included and integrated (starting from 3,855, only 25 were finally included).
Among the aspects to be reduced, the abstract could be significantly reduced, paragraph 4. Discussion is duplicated and the second paragraph should be eliminated, and the sections recommendations, future directions and conclusions could be reduced and unified in a single section. The references on the last page have not been properly formatted, especially in the name of the sources.
